# Can Gratitude Ease the Burden of Fibromyalgia? A Systematic Review

**DOI:** 10.3390/bs15081079

**Published:** 2025-08-07

**Authors:** Bruno Daniel Carneiro, Daniel Humberto Pozza, Isaura Tavares

**Affiliations:** 1Unit of Experimental Biology, Department of Biomedicine, Faculty of Medicine, University of Porto, 4200-319 Porto, Portugal; dhpozza@gmail.com (D.H.P.); isatav@med.up.pt (I.T.); 2Rheumatology Service, Unidade Local de Saúde do Alto Minho, Hospital Conde de Bertiandos, 4990-078 Ponte de Lima, Portugal; 3Institute for Research and Innovation in Health and IBMC, University of Porto, 4200-135 Porto, Portugal

**Keywords:** chronic pain, fibromyalgia, acceptance, coping, quality of life, psychological well-being, symptom management, emotional health, functional outcomes, cognitive–behavioral therapy, mindfulness, chronic illness

## Abstract

Fibromyalgia has unclear etiopathogenesis, no curative treatment, and a severe impact on the quality of life. Gratitude practices have been shown to enhance the quality of life in chronic diseases. This systematic review, performed by searching five electronic databases, following the PRISMA guidelines, is the first aiming to evaluate the impact of gratitude in fibromyalgia. Data from eligible studies was extracted and a narrative synthesis was performed. Six articles (four observational studies and two randomized clinical trials) were included. Higher levels of gratitude are associated with reduced symptom severity, an enhanced quality of life, improved well-being, and the improvement of pain-related outcomes in fibromyalgia patients. Gratitude is related to reduced stress, anxiety, and depression; better sleep patterns; and less functional impairment in FM patients. Higher levels of gratitude contribute to a better quality of life, general well-being, and higher functioning capacity in fibromyalgia patients. Based on the results gathered in this systematic review, we propose that gratitude should be investigated as a therapeutic adjuvant in the management of fibromyalgia.

## 1. Introduction

Fibromyalgia (FM) is a complex and challenging clinical entity with a peak of incidence occurring between 50 and 60 years of age ([42]; [43]; [64]; [90]). The prevalence of FM varies with the methods of study and differs geographically ([1]; [13]; [19]; [32]; [58]; [63]; [64]; [70]; [77]; [79]; [86]; [89]; [90]). In many European countries (Portugal, Spain, France, Germany, and Italy), the prevalence of FM ranges from 2.9% to 4.7% ([75]). It is higher in the United States of America (USA) (6.4%) and reaches 9.3% in Tunisia ([75]). FM emerges as the third most common prevalent musculoskeletal condition after lumbar pain and osteoarthritis ([80]). The disease is more prevalent in females (female-to-male ratio: 3-1; [64]).

Fibromyalgia is a chronic pain disorder characterized by widespread pain, paresthesias described as pins-and-needles or a tingling sensation, fatigue, and sleep disturbances ([8]; [30]; [46]; [65]; [73]). Symptoms vary in type, severity, and location, being influenced by factors like stress and other comorbidities ([12]; [17]; [47]; [68]; [81]). FM is also linked to negative emotions ([28]) and may co-occur with increased stress, anxiety, and depression disorders ([31]).

The diagnosis of FM is primarily clinical ([75]). Physical examination has limited reproducibility and validity, making a thorough clinical history-taking crucial for gathering diagnostic elements ([36]). One of the challenges in diagnosing and treating FM is the unknown pathophysiology of the disease ([21]; [50]). Two main hypotheses have been proposed. The first suggests that immune and inflammatory alterations cause the sensitization of the central and peripheral nervous systems in FM, as evidenced by increased pro-inflammatory cytokines in plasma and in cerebrospinal fluid and decreased anti-inflammatory cytokines in blood ([6]; [51]; [67]; [87]). The second hypothesis proposes that FM is a central sensitivity syndrome, characterized by the heightened sensitization of the somatosensory system, leading to amplified nociceptive signals and increased pain perception ([18]; [40]; [88]). Additionally, cognitive and emotional sensitization in FM involves negative cognitive bias, rumination, and catastrophization ([14]). More recently, it was proposed that FM results from an interplay between genetic predisposition and stressful events, which affects the peripheral and central nervous system ([75]). Regarding the mechanisms underlying the relationship between stress and pain in FM patients, alterations in cortisol levels have been suggested as a potential link between low stress tolerance and the amplification of pain ([66]). Patients with FM experience nociplastic pain, which reflects complex interactions between the central nervous system and the body periphery. This involves both bottom–up (primarily inflammatory and algesic) and top–down (primarily psychological and cognitive–emotional) processes. These nociplastic and pathogenic mechanisms work together to perpetuate the clinical manifestations of the disease and account to its chronicity ([75]).

A holistic management of FM has been proposed ([75]), considering not only on the symptoms experienced by the patients but also the factors that can alleviate or aggravate these symptoms. Additionally, the evaluation of FM patients should address the impact of FM on daily life, functional status, and working ability ([75]). FM can be assessed using validating tools such as the Fibromyalgia Impact Questionnaire (FIQ) ([16]) and its revised version ([7]; [71]), the Fibromyalgia Assessment Status (FAS) ([41]; [72]), the Fibromyalgia Survey Criteria (FSC) ([35]), and the Patient Health Questionnaire 15 (PHQ15) ([34]). However, some controversy exists regarding the best methods to diagnose FM, and the absence of specific biomarkers remains the greater gap and barrier in the diagnosis ([75]), with research in this field still in its early stages ([4]). Currently, the two most up-to-date sets of diagnostic criteria for FM are the 2016 revisions to the 2010/2011 American College of Rheumatology (ACR) FM diagnostic criteria ([91]) and the ACTTION (Addiction Clinical Trial Translations Innovations Opportunities and Networks)-APS (American Pain Society) Pain Taxonomy (AAPT) core diagnostic criteria for FM ([3]).

In the holistic management of FM, a combination of pharmacological and non-pharmacological therapies is recommended. Key strategies include patient education, physical exercise, and a combination of medications like antidepressants, anticonvulsants, and other pain relievers ([52]). However, there is no gold standard pharmacological therapy to FM ([75]). Psychotherapy, especially cognitive–behavioral therapy (CBT), is effective for managing mood disorders and maladaptive coping ([11]; [29]). In the latter, mindfulness-based interventions have been proposed, along with other approaches directed to the emotional components of the disease ([48]; [50]; [62]).

As a chronic pain syndrome, FM can significantly compromise the quality of life, in what concerns physical, cognitive, social, and emotional aspects of life ([9]; [44]), with a huge burden both in psychological and mental health ([57]). Cultivating positive psychological traits has been conceptually proposed as a possibility to improve mental health and the quality of life and decrease fatigue in FM patients ([27]; [39]; [97]). Gratitude is the main positive psychological trait that is fostered in some mindfulness-based interventions ([50]). In fact, gratitude, meaning noticing and appreciating the positive aspects of life, whether they are tangible or intangible ([74]; [96]), is a growing area of research. The neural mechanisms underlying the effects of gratitude are under study and may include alterations in the activity of the medial prefrontal and anterior cingulate cortices ([26]; [45]). Gratitude has been linked to improved sleep quality and lower levels of depression in healthy populations ([94], [92]), showing higher well-being ([95]) and higher adaptative coping mechanisms ([93]). Practices focusing on gratitude can improve the quality of life in patients with chronic health conditions, namely heart failure ([56]), arthritis ([24]), chronic obstructive pulmonary disease ([24]), diabetes ([24]), or breast cancer ([69]). In fact, in the field of oncology, the use of gratitude receives preponderant attention with a continuous increase in scientific production ([33]). Recent advances in the field of gratitude practices for oncology patients should be highlighted ([2]; [5]; [10]; [83]). Recently, it was shown that targeting gratitude as a protective resource may be a potential therapeutic tool to improve psychological and behavioral outcomes in older adults with chronic low back pain ([53]).

Based on the abovementioned results, it is possible that gratitude could ease the burden of living with FM. However, maintaining a sustained attitude of gratitude can be challenging for FM patients due to the complexity of the symptoms in this chronic disease ([85]). Since there is no robust evidence to establish the prescription of gratitude to FM patients, we performed, for the first time, a systematic review aiming to evaluate the impact of gratitude on the quality of life, well-being, and functional outcomes in patients with FM. The review synthesizes the current evidence on the associations between gratitude and FM symptoms, including pain, fatigue, sleep disturbances, and emotional health (stress, anxiety, and depression), to identify potential therapeutic benefits and inform future research and clinical practice.

## 2. Materials and Methods

This systematic review adheres to the Preferred Reporting Items for Systematic Reviews and Meta-Analyses (PRISMA) 2020 guidelines ([61]). We follow the PROSPERO database protocol model to ensure study transparency and reproducibility (ID: CRD42024565556). The review question was “In human individuals with fibromyalgia (P), how does cultivating gratitude (I) compared to standard care or other interventions not involving gratitude (C) impact the experience of fibromyalgia in terms of pain and other outcomes (O)?”.

A search in five electronic bibliographic databases, which included Web of Science, PubMed, Scopus, CENTRAL (Cochrane Central Register of Controlled Trials), and PsycInfo, was carried out on 31 January 2025. The search strategy was built up combining the terms “fibromyalgia” and “gratitude” and using the Boolean operator “AND” as follows: (1) Web of Science—“TS = (fibromyalgia AND gratitude)”; (2) PubMed—“(“Fibromyalgia”[MeSH Terms] OR fibromyalgia[Title/Abstract]) AND (gratitude[Title/Abstract])”; (3) Scopus—“TITLE-ABS-KEY(fibromyalgia AND gratitude)”; (4) CENTRAL—“(fibromyalgia):ti,ab,kw AND (gratitude):ti,ab,kw”; (5) PsycInfo—“(fibromyalgia AND gratitude).ti,ab,kw.” A citation search was also performed based on key articles identified in the main search and using the tool “cited by”.

The included studies had to (1) include the terms “fibromyalgia” and “gratitude”, at least in the titles and/or abstracts and/or keywords; (2) involve participants diagnosed with FM; (3) involve interventions explicitly designed to promote gratitude or gratitude-related activities not only explicitly designed to promote only gratitude; (4) be classified as clinical trials or observational studies; (5) be published in peer-reviewed journals; and (6) involve participants aged 18 years and older. We opted for exclude studies (1) that focused solely on acute pain or did not focus on FM; (2) where having gratitude and/or gratitude practices was not the primary focus or intervention; (3) that were in non-English languages; (4) that were editorials, conference abstracts, and review articles; or (5) that involved animals.

Screening of titles and abstracts of identified records was carried out independently by two reviewers to determine eligibility in the Rayyan tool (https://rayyan.ai/). The “blind mode” was on to ensure that reviewers were not influenced by each other’s decisions, thereby maintaining objectivity and minimizing bias in the screening process. Full texts of potentially relevant studies were then assessed for a comprehensive analysis. A third member of the research team helped solve any conflicts. The level of agreement between the authors was assessed using the Kappa test ([55]).

Data from eligible studies was extracted. The information collected from each study included information about authors, publication year, publication country, type of study, study characteristics, participant characteristics, intervention, control, measures of effect, limitations, main results, and implications for future research. The risk of bias of the clinical trials was evaluated with the Cochrane RoB 2 tool ([82]) at the outcome level visualized with the Cochrane risk of bias VISualization app 4.0 ([54]) and the risk of bias of other study types was evaluated with the ROBINS-E tool ([37]), also at the outcome level visualized with the Cochrane risk of bias VISualization app 4.0 ([54]).

A narrative synthesis was conducted to summarize the findings of the included studies. Given the number of included studies and their characteristics, it was decided to present a qualitative synthesis and not a meta-analysis. Also, because of this, we highlight that some deviations from the initial registered protocol were needed, namely in terms of the inclusion criteria, the definitions of outcomes, and the rationale of the study.

## 3. Results

A comprehensive literature search in the five databases and in citation searching resulted in 69 potential records identified. Following the removal of duplicate records, fifty-five manuscripts remained for the title and abstract review (fifty-one articles identified via databases and four articles identified via citation searching). Of the articles selected via databases, a total of 42 records were excluded after screening by title and abstract. All the articles identified via citation searching were excluded during the screening by title and abstract. So, nine articles were selected for full-text examination. The final inclusion criteria were met by six studies (initial identified via databases). A concordance index of 94% (Kappa coefficient of 0.94) was obtained between the authors, indicating an almost-perfect agreement between the reviewers. Figure 1 shows the entire search process, including the reasons for manuscript exclusion during the screening process. The summary of findings is shown in Table 1.

### 3.1. Characteristics of Included Studies

Of the selected six articles, two studies were randomized controlled clinical trials (RCT) ([60]; [99]), three were cross-sectional studies ([20]; [38]; [85]), and one was a qualitative study ([76]). All studies tried to establish a relation between gratitude and different outcomes associated with FM.

Of the included studies, three studies resulted from the collaboration of authors from a single country, applicable to the USA ([60]; [76]) and Brazil ([99]), and three studies resulted from the collaboration of authors from different countries, namely the USA, United Kingdom (UK), Germany, Austria, and Poland ([20]; [38]; [85]). All studies involved both women and men, and the number of female patients was higher (545 females and 179 males, with a female-to-male ratio of 3-1). In the RCT, there were eighty-nine women and six men ([60]) and thirty-nine women and one man ([99]). In the cross-sectional studies, there were one-hundred-and-sixty-one women and nine men ([85]), one-hundred-and-fifty-two women and one-hundred-and-twenty-six men ([38]), and ninety-seven women and thirty-six men ([20]). The qualitative study ([76]) included seven women and one man.

Both RCTs compared an intervention involving gratitude with an active control. A comparison between a mobile app that promotes the practice of gratitude over 6 weeks (intervention group) with a paper booklet (control) was performed to evaluate the health-related quality of life through questionnaires about FM’s impact, symptoms, pain, and self-care ([99]). A comparison between online teaching over 5 weeks and using gratitude to foster positive affect and reduce stress, pain, and fatigue (LARKSPUR) with completing daily emotional reports was performed to evaluate pain intensity and pain interference, pain catastrophizing, system–physical function, system fatigue, and the daily positive and negative affect in emotions ([60]).

The three cross-sectional studies evaluated different outcomes. Toussaint and collaborators measured the levels of gratitude, anxiety, and depression symptoms and the quality of life with questionnaires ([85]). Hirsch and collaborators measured the levels of gratitude, self-compassion, self-forgiveness, stress, anxiety, depressive symptoms, sleep quality, and functional impairment ([38]). In this study, regular clinic attendees, originating from Austria, Germany, and Switzerland answered an anonymous online survey administered via a secure server. Charzyńska and collaborators evaluated the levels of life satisfaction, health status, depression, anxiety, stress, fatigue, self-forgiveness, and forgiveness to others and gratitude ([20]). In this study, patients regularly seeking treatment at the Gastein Healing Gallery in Bad Gastein, Austria, answered an anonymous online survey administered via a secure server.

In the qualitative study ([76]), Caucasian patients from the Midwest of the USA were interviewed over 1 to 3 h and the transcripts were screened to find components associated with the process of “transformation to a more authentic self”, with emphasis on the expression of gratitude.

### 3.2. Summary of Key Findings

Regarding the results of the RCT conducted by Yuan and collaborators, FM patients allocated to the intervention group showed more relevant improvements in FM symptom severity and in the quality of life ([99]).

Ong and collaborators demonstrated that the intervention group showed significantly improvements in positive affect and negative affect in comparison with the baseline and post intervention, which were maintained for 1 month ([60]). Furthermore, when compared to the control group, the improvements were more substantial ([60]). The LARKSPUR group also showed significant and sustained reductions in PCS (pain catastrophizing scale) and PROMIS-SI (patient-reported outcome measurement information system—pain intensity), along with greater improvements in PROMIS-PF (patient-reported outcome measurement information system—physical function) and PROMIS F-SF (patient-reported outcome measurement information system fatigue—short form), compared to the control group ([60]).

In two of the cross-sectional studies, it was shown that FM patients had lower levels of gratitude than healthy controls ([38]; [85]). The other cross-sectional study showed that FM patients had lower levels of gratitude than patients with arthritis ([20]). A positive association between gratitude and the quality of life in FM patients was identified ([85]). The lower quality of life of FM patients in relation to healthy controls was proposed to be due to lower levels of gratitude ([85]). In fact, gratitude appeared to be generally related to reduced stress, anxiety, and depression, which, in turn, were related to better sleep patterns and less functional impairment ([38]). This study also highlighted that a gratitude-focused model explained about 25% of the total variance in FM patients’ functional impairment ([38]). Along with this, gratitude was negatively associated with anxiety, depression, stress, and disability and was positively related with sleep quality, self-compassion, and self-forgiveness ([38]). In the study of Charzyńska and collaborators, it was shown that patients with the profile “life dissatisfaction” and profile “suboptimal well-being” had lower levels of gratitude and higher proportion of FM diagnosis ([20]).

The qualitative study identified ten central components that were necessary to FM patients to achieve a transformation into “a more authentic self” that may have allowed FM patients to live “fully functioning lives”, resuming “full life activity free of pain”. In this study, expressing gratitude was highlighted as one of the ten analyzed components ([76]).

### 3.3. Risk of Bias

The graphical representations of the risk of bias of the analyzed studies are shown in Figure 2 and Figure 3. Regarding the RCTs (Figure 2), one study revealed some concerns ([99]) and one study was classified as having a high risk of bias ([60]). Concerning the observational studies (Figure 3), two revealed some concerns ([38]; [85]) and two were classified as having a high risk of bias ([20]; [76]).

## 4. Discussion

To the best of our knowledge, this is the first systematic review analyzing the effects of gratitude on various aspects of the lives of patients with FM, namely symptom severity, the quality of life, and overall well-being. All the studies included in this review demonstrated benefits of gratitude practices in FM patients.

Expressing gratitude is probably one of the characteristics necessary for FM patients to become more authentic selves and may be associated with resuming “functioning lives” with “activity free of pain” ([76]). This is in line with the conclusions of a recent study that highlighted the fact that gratitude and cognitive empathy may promote a feeling of “being authentic” ([84]). Being grateful and being authentic could functioning as positive individual traits that might have protective effects in human health ([78]). Positive individual traits can improve the quality of life and prevent pathologies that arise when life is barren and subjectively perceived as meaningless ([78]). This has a particular impact on FM, which is frequently accompanied by depressive signs ([31]) and where patients live with daily pain ([75]), with a negative functional impact ([9]; [44]).

Acts of gratitude can be used as a complementary therapeutic approach to treat anxiety and depression ([23]) and gratitude increases positive feelings and emotions in the general population ([23]) with positive impacts on mental and physical well-being ([93], [95]). Gratitude has also benefits in chronic pain conditions such as chronic low back pain ([53]) and is associated with higher sleep quality and lower levels of depression in healthy populations ([94], [92]) and with higher well-being in general population ([95]). In a population of 224 chronic pain patients, gratitude was shown to have a direct positive effect in depression ([59]). This was in line with studies performed in larger populations (761 Korean adults) showing that gratitude is an essential element to improve subjective well-being, which involves life satisfaction, hope, and positive affect and negative affect ([98]). It is important to address affect as positive affect encompasses emotions like attentiveness, determination, and interest while negative affect includes emotions such as fear, sadness, and anger. Both affects are biologically rooted temperament dimensions that play a key role in the onset and persistence of anxiety and depression, particularly in cases where a patient has high negative affect and low positive affect stability ([15]; [100]). The results of one of the studies analyzed in this systematic review are clearly in line with these overall results. The study of Toussaint and collaborators showed a positive association between gratitude and the quality of life in FM patients and proposed that the difference in the quality of life between FM patients and healthy controls can be partially explained by levels of gratitude ([85]). Another analyzed study also showed that a gratitude-focused model could explain a quarter of the total variance in functional impairment among FM patients ([38]). This impact of gratitude on the quality of life of FM patients may be explained by the association between higher levels of gratitude in FM patients with lower levels of anxiety, depression, stress, and disability and with higher levels of sleep quality, self-compassion, and self-forgiveness ([38]). Interestingly, in an analyzed study where other painful musculoskeletal diseases were included, namely arthritis, patients with FM had lower levels of gratitude ([20]), indicating that it is not only pain per se but the complex features of FM that could be improved by gratitude practices. Collectively, the results of the cross-sectional studies are in line with the results of other diseases and support our initial hypothesis that gratitude could be associated with a better quality of life and better functioning in FM patients.

According to the RCTs included in the review, gratitude-based interventions have a positive impact on FM patients’ outcomes, further reinforcing our initial hypothesis. FM patients allocated to gratitude-based interventions had more relevant improvements in symptom severity ([99]), the quality of life ([99]), and in both positive affect and negative affect ([60]). The fact that gratitude has a positive influence in both positive affect and negative affect is aligned with the observation that patients with higher levels of gratitude have lower levels of depression ([38]). Moreover, FM patients allocated to the intervention group showed significant and sustained reductions in PCS and PROMIS-SI ([60]) and greater improvements in PROMIS-PF and PROMIS F-SF ([60]), which were associated with a better quality of life and better functioning. These results agree with the data from another RCT that evaluate the effect of a gratitude intervention on a community sample of adults in relation to aspects involving well-being and mental health ([22]) and the results obtained in a series of classic quasi-experiments that studied the effects of rumination, distraction, and gratitude on positive affect and negative affect ([49]). It was emphasized that gratitude interventions increase positive affect, subjective happiness, and life satisfaction and reduce negative affect and depression symptoms ([22]). Furthermore, it was highlighted that gratitude condition promoted positive affect when compared to rumination and distraction conditions ([49]). Interestingly, a previous study had already emphasized that gratitude-outlook groups exhibited heightened well-being ([25]).

Overall, the data summarized in this review suggests a relationship between FM and gratitude. Gratitude appears to positively impact on FM patients’ outcomes by enhancing their quality of life and physical and mental well-being, thereby improving their functional capacity. Future research should explore gratitude as a potential therapeutic target for patients with FM. Integrating personalized gratitude practices alongside primary treatments could be valuable as complementary therapies may reduce reliance on pharmacological interventions and their associated side effects ([62]).

Despite the promising results obtained, the interpretations and conclusions of this systematic review should be made with caution. Regarding the female-to-male ratio, and in spite of the fact that it represented, exactly, the 3-1 ratio in the prevalence of the disease ([50]; [64]), the findings from the analyzed studies may be more representative of the female experience with FM. It is important to consider this when interpreting the results and generalizing them to the broader population. This review has not included a meta-analysis, and has encompassed only include six studies, of which only two were RCTs, and had some limitations. Some limitations could be identified in the studies, namely small sample sizes ([60]; [76]; [99]), small periods of intervention ([60]; [99]), and the characteristics of the included populations, such as the fact that respondents were mostly older adults ([38]) and the fact that most of the patients were more open to complementary medicine ([76]). Four studies ([20]; [38]; [60]; [99]) did not adjust the results for confounding variables. In the study carried out by [76] ([76]), snowball sampling could have influenced the results. In line with this, the convenience sample recruited from a patient self-help group used by [85] ([85]) and the volunteers recruited from a spa gallery used by [20] ([20]) also had limitations due to the convenience sampling method. In the RCTs ([60]; [99]), interventions did not include only the practice of gratitude. Therefore, the direct effect of gratitude was difficult to measure. Since most of the effect measures used by the different authors involved patients answering surveys or patients’ interviews, there may have been an associated response bias. Furthermore, some of the included studies used the diagnosis of FM based on physical examination and/or the 2010/2011 ACR diagnostic criteria ([91]), with some studies relying on self-reported diagnoses. More recent diagnostic criteria are available, which may influence the findings ([3]; [91]). In spite of the limitations that impair generalizing the results obtained in this systematic review, to the best of our knowledge, this was the first study that addressed FM and gratitude with the primary objective of evaluating the impact of gratitude on the quality of life, well-being, and functional outcomes in patients with FM, which is of great value to the scientific community and for FM patients.

Future research in this field should prioritize developing an RCT that allows to fully evaluate the evidence that gratitude positively impacts the quality of life, the well-being, and the functioning of FM patients. The future design of studies using gratitude in FM management should be critically discussed with more objective measures to determine the mechanisms underlying the beneficial effects of gratitude. For example, performing dose–effect studies and evaluating stress by measuring cortisol levels could provide important results. With FM being a common clinical entity all over the world ([75]), the behavioral practices of the patients need to be better explored in the therapeutic approach to this syndrome to provide some hope to FM patients.

## 5. Conclusions

Based on the promising results gathered in this systematic review, we reinforce our proposal that gratitude should be investigated as a therapeutic adjuvant in the holistic management of FM. This systematic review underscores the importance of continued research to translate these findings into a more effective management of FM. The directions of research should focus on the underlying mechanisms and on the definition of adequate experimental design. Meanwhile, for the mechanisms, brain imaging studies may be relevant, and in the experimental design, the importance of RCTs with adequate active controls should be considered. Including gratitude practices may foster a more humanistic approach to manage FM that focuses on the patient’s subjective experience and empowers patients to embrace several aspects of their lives besides the disease; foster the interpersonal relationships, namely with their relatives and clinical staff; and ultimately improve their overall quality of life, even in the face of the disease.

## Figures and Tables

**Figure 1 behavsci-15-01079-f001:**
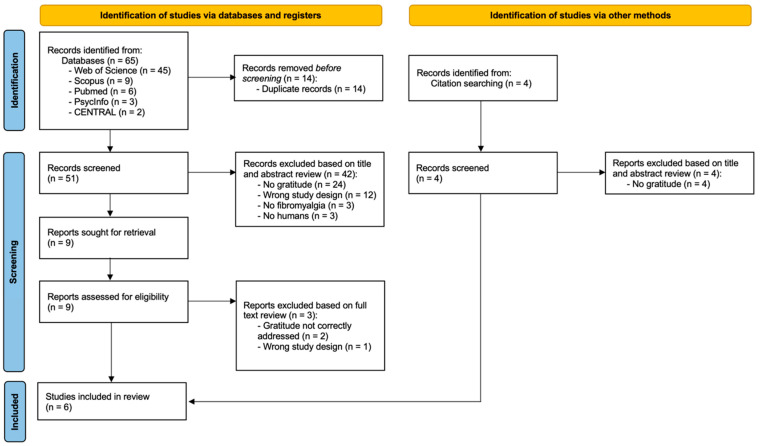
PRISMA flow diagram outlining the selection of the included studies.

**Figure 2 behavsci-15-01079-f002:**
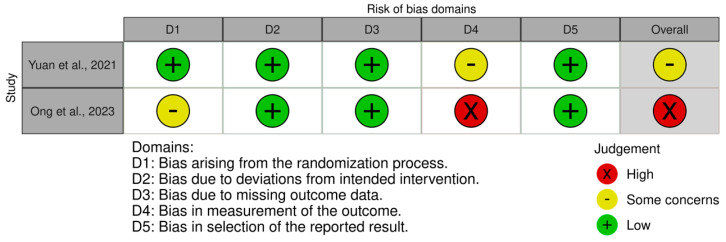
Risk of bias, represented in five categories and with its overall result, for each randomized clinical trial included in the review ([99]; [60]).

**Figure 3 behavsci-15-01079-f003:**
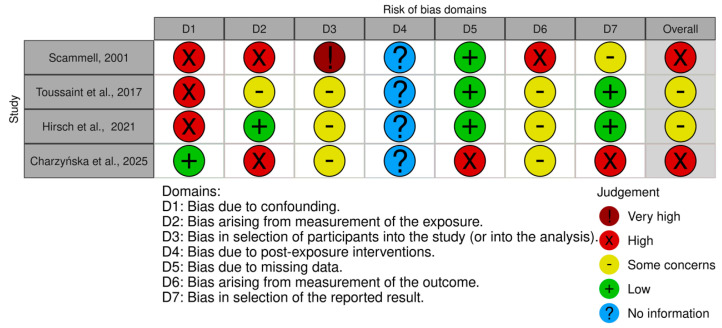
Risk of bias, represented in seven categories and with its overall result, for each observational study included in the review ([76]; [85]; [38]; [20]).

**Table 1 behavsci-15-01079-t001:** Comprehensive overview directed towards gratitude in the key characteristics of the included studies, organized in chronologic order.

Reference Study Type Country(ies) *	Participant Characteristics	Intervention	Control	Measures of Effect	Main Results and Implications for Future Research	Limitations
[76] ([76]) Qualitative study USA	- 8 Caucasian subjects (7 women + 1 man) all officially diagnosed with FM that had experienced a positive transformation through the experience of dealing with FM - mean age of 47 years	Observational	Observational	- interviews were coded (using open coding techniques) to find the components associated with the “transformation to a more authentic self”	- 10 components of change emerged as central to a “transformation to a more authentic self” in subjects with FM, one of them being “expresses gratitude”	- small sample size - snowball sampling - the subjects’ sense of how to answer may have been influenced by the researcher’s presence, holding her own point of view
[85] ([85]) Cross-sectional study Austria Germany UK USA	- 170 subjects with FM diagnosis: mean age of 58 years; 161 female subjects + 9 male subjects - 81 healthy controls: mean age of 47 years; 76 female subjects + 5 male subjects	Observational	Observational	- gratitude (with GQ6) - anxiety and depression symptoms (with HADS) - quality of life (with QoLS) - health-related quality of life (with SF12)	- FM subjects had lower levels of gratitude in comparison to healthy controls - gratitude had a positive association with quality of life - the difference between FM subjects and healthy controls in quality of life was partially explained through gratitude	- causal direction cannot be inferred - convenience sample - all measures were self-reports - confounding variables not controlled
[38] ([38]) Cross-sectional study Austria Germany UK USA	- 1218 subjects (632 women + 586 men) with different rheumatic and musculoskeletal diseases - mean age of 58 years - 278 subjects with FM diagnosis (152 women + 126 men)	Observational	Observational	- gratitude (with GQ6) - self-compassion (with SCS) - self-forgiveness (with SFFOI) - stress (with PSS4) - anxiety (with GADS) - depressive symptoms (with PHQ2) - sleep quality (with SCI) - functional impairment (with HAQ)	- gratitude was significantly and negatively, although weakly, related to anxiety, depression, stress, and disability - gratitude was significantly and positively, and weakly to moderately, related to sleep quality, self-compassion, and self-forgiveness - gratitude was related to reduced stress, anxiety, and depression in parallel, which were in turn related to better sleep quality and less functional impairment	- potential self-selection sampling bias preclude examination of causality - medical or psychiatric treatment, comorbid disease, and the presence of social support may impact functional ability - respondents were mostly older adults
[99] ([99]) Randomized clinical trial Brazil	- 40 subjects with FM diagnosis (39 women + 1 man) randomized into ProFibro group (20) and control group (20) - age between 19 and 59 years	Mobile app ProFibro for 6 weeks (this app promotes the practice of gratitude and provides educational measures regarding FM)	Paper booklet of similar content to the content of the ProFibro app for 6 weeks	- health-related quality of life (with FIQR) - symptoms (with WPI, VAS and SSS) - self-care agency (with RASCAS)	- ProFibro group showed more relevant improvements in symptom severity and in quality of life	- small sample size - short period of intervention - intervention did not included only the practice of gratitude so the direct effect of gratitude is difficult to measure - confounding variables not controlled
[60] ([60]) Randomized clinical trial USA	- 95 subjects with FM diagnosis (89 women + 6 men) - age between 50 and 80 years	LARKSPUR group (49 subjects) for 5 weeks (this program included skill training that targeted 8 skills to help foster positive affect; one of them was gratitude)	Emotion reporting group (46 subjects)	- pain intensity and pain interference (with PROMIS-SI and PROMIS-PI) - pain catastrophizing (with PCS) - physical functioning (with PROMIS-PF and PROMIS F-SF) - daily positive and negative affect (with mDES)	- the LARKSPUR group showed significant improvements in both positive and negative affect from baseline to post-intervention, with these gains sustained at the 1-month follow-up - LARKSPUR led to greater improvements in positive/negative affect and reductions in pain-related outcomes (PCS, PROMIS-SI, PROMIS-PI) compared to the control group, maintained at the 1-month follow-up - while the LARKSPUR group experienced significant reductions in pain catastrophizing, these improvements were not fully sustained at the 1-month follow-up	- small sample size - short period of intervention - intervention did not included only the gratitude so the direct effect of gratitude was difficult to measure - study relied on self-report of FM diagnosis - confounding variables not controlled
[20] ([20]) Cross-sectional study Austria Germany Poland UK USA	- 133 subjects with FM (97 women and 36 men) - mean age of 58 years	Observational	Observational	- life satisfaction (with a single question: “All things considered, how satisfied are you with - your life as a whole?”) - health status (with EQ-5D) - depression and anxiety (with PHQ4) - stress (with PSS4) - fatigue (with GF) - self-forgiveness and forgiveness to others (with TFS) - gratitude (with GQ6)	- patients with profile “life dissatisfaction” and profile “suboptimal well-being” had lower levels of gratitude and higher proportion of FM diagnosis - patients with FM had lower levels of gratitude compared with patients with arthritis	- causal direction cannot be inferred - convenience sample - potential response bias - confounding variables not controlled

Legend: * Based on all authors’ affiliation. Abbreviations—EQ-5D, euroqol five dimensions; FIQR, revised fibromyalgia impact questionnaire; FM, fibromyalgia; GADS, generalized anxiety disorder scale; GF, general fatigue subscale; GFPA, German fibromyalgia patient association; GQ6, gratitude questionnaire 6; HADS, hospital anxiety and depression scale; HAQ, health assessment questionnaire; LARKSPUR, lessons in affect regulation to keep stress and pain under control; mDES, modified differential emotions scale; PCS, pain catastrophizing scale; PHQ2, patient health questionnaire 2; PHQ4, patient health questionnaire 4; PROMIS F-SF, patient-reported outcome measurement information system fatigue—short form; PROMIS-PF, patient-reported outcome measurement information system—physical function; PROMIS-PI, patient-reported outcome measurement information system—pain interference; PROMIS-SI, patient-reported outcome measurement information system—pain intensity; PSS4, perceived stress scale 4; QoLS, quality of life scale; RASCAS, revised appraisal of self-care agency scale; SCI, sleep condition indicator; SCS, self-compassion scale; SF12, short form 12; SFFOI, self-forgiveness and forgiveness of others index; SSS, symptom severity scale; TFS, Toussaint forgiveness scale; UK, United Kingdom; USA, United States of America; VAS, visual analogue scale; WPI, widespread pain index.

## Data Availability

All data generated or analyzed during this study are included in this article.

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
