# Peer review of "Can Gratitude Ease the Burden of Fibromyalgia? A Systematic Review"

_behavsci, 2025, doi:10.3390/bs15081079_

Round 1
Reviewer 1 Report
Comments and Suggestions for Authors
Dear authors,
Please accept my appreciation for your thoughtful review article. The topic is highly relevant. Indirectly, it also highlights the need to better understand the pathophysiological mechanisms of gratitude and its potential clinical applications.
I would, however, kindly suggest revising the statement "when life is meaningless" in your discussion section. Perhaps rephrasing it as "when life is subjectively perceived as meaningless" would be more appropriate. Meaning in life is a deeply subjective phenomenon and can be experienced as fulfilling even under unimaginable circumstances. This phrasing might otherwise unintentionally disregard the resilience of patients who continue to experience meaning despite extreme limitations. A poignant example is the phenomenon of reported happiness in patients with locked-in syndrome, though methodological concerns remain (cf. Bruno MA, Bernheim JL, Ledoux D, et al., BMJ Open, 2011;1(1):e000039).
I would also encourage you to more explicitly address the potential effects of gratitude in oncological patients (cf. Giving and receiving thanks: a mixed methods pilot study of a gratitude intervention for palliative patients and their carers. BMC Palliative Care. 2023;22:52).
Further elaboration on potential study directions and therapeutic implementation of gratitude-based interventions might be inspiring to readers. For instance, as a clinician, I have observed that—when delivering difficult diagnoses such as glioma—I intuitively try to evoke a sense of gratitude in patients for what remains: a loving family, preserved function, or moments of clarity. Anecdotally, this often leads to a noticeable shift in emotional tone...
It would be fascinating to see gratitude-based interventions studied even more systematically in neuro-oncological populations. As we explore such deeply subjective and neurobiologically complex phenomena, we remain aware of the epistemological paradoxes they pose—perhaps akin to attempting to understand consciousness with the very organ that generates it.
Thank you again for your valuable contribution.
Author Response
Dear authors,
Please accept my appreciation for your thoughtful review article. The topic is highly relevant. Indirectly, it also highlights the need to better understand the pathophysiological mechanisms of gratitude and its potential clinical applications.
Answer: We acknowledge your comment. Thank you for your recognition. Based on your comment, we added a sentence about a future trend of research in the mechanisms of gratitude which are related to the effects of gratitude at the prefrontal and anterior cingulate cortices. Please see lines 110-112 at the “Introduction”.
- I would, however, kindly suggest revising the statement "when life is meaningless" in your discussion section. Perhaps rephrasing it as "when life is subjectively perceived as meaningless" would be more appropriate. Meaning in life is a deeply subjective phenomenon and can be experienced as fulfilling even under unimaginable circumstances. This phrasing might otherwise unintentionally disregard the resilience of patients who continue to experience meaning despite extreme limitations. A poignant example is the phenomenon of reported happiness in patients with locked-in syndrome, though methodological concerns remain (cf. Bruno MA, Bernheim JL, Ledoux D, et al., BMJ Open, 2011;1(1):e000039).
Answer: We fully agree and corrected the sentence (please see section “Discussion”, line 324).
- I would also encourage you to more explicitly address the potential effects of gratitude in oncological patients (cf. Giving and receiving thanks: a mixed methods pilot study of a gratitude intervention for palliative patients and their carers. BMC Palliative Care. 2023;22:52).
Answer: We highlighted cancer as a disease in which gratitude practices are studied in more detail. Please see the “Introduction” in lines 119-123. Besides this reference, we added 3 additional references, along with a recent bibliometric study in the field of gratitude and cancer.
- Further elaboration on potential study directions and therapeutic implementation of gratitude-based interventions might be inspiring to readers. For instance, as a clinician, I have observed that—when delivering difficult diagnoses such as glioma—I intuitively try to evoke a sense of gratitude in patients for what remains: a loving family, preserved function, or moments of clarity. Anecdotally, this often leads to a noticeable shift in emotional tone...
Answer: We deeply acknowledge this inspiring information about a very important and humanistic medical practice. Based on your observation, and also in question 5 of reviewer 2, we changed considerably the “Conclusions” section (please, see the section).
- It would be fascinating to see gratitude-based interventions studied even more systematically in neuro-oncological populations. As we explore such deeply subjective and neurobiologically complex phenomena, we remain aware of the epistemological paradoxes they pose—perhaps akin to attempting to understand consciousness with the very organ that generates it.
Thank you again for your valuable contribution.
Answer: We acknowledge your comment. We share with you this fascination about interventions that use gratitude, namely in this challenge syndrome, the FM. However, and as outlined in our manuscript, sometimes gratitude is included in other approaches, namely in Mindfulness-based interventions. There are very interesting studies in the field of neurooncology (e.g., Milbury K et al., 2020; J Pain Symptom Manage; doi: 10.1016/j.jpainsymman.2020.02.004). There are also touching examples on how patients with brain tumors express gratitude in this magnificent study 'Make Them Wonder How You Are Still Smiling': The Lived Experience of Coping With a Brain Tumour (Zanotto et al., 2023; doi: 10.1177/10497323231167345). Nevertheless, a causal relation between being grateful and having a better management of the disease cannot be established. Finally, in another qualitative study of patients with brain tumors evaluating the type of Reddit user posting by patients with brain tumors, the expression of gratitude also emerges. Overall, there are some studies about gratitude in neuro-oncology but this field needs to be considerably expanded. Thank you for your magnificent example as clinician.
Reviewer 2 Report
Comments and Suggestions for Authors
The work is appropriate for the journal and of interest to readers.
Title and Abstract:
The title is clear and adequately reflects the content of the study.
The abstract covers the fundamental aspects: justification, method, results and conclusion. However, it uses vague phrases such as “Gratitude appears to be related...” that do not reflect the level of evidence derived from the study. In addition, it mentions 6 included studies but does not distinguish between types of studies (clinical vs. observational trials).
Introduction:
The writing is quite dense and has excessive citations that interrupt the flow.
Some citations are outdated or overused (e.g., Sarzi-Puttini et al., 2020 appears more than 10 times).
Methodology:
The PRISMA 2020 protocol is followed, recorded in PROSPERO and the PICO question is set.
However, the complete search strategy is not detailed, nor is the table of MeSH terms or descriptors provided. Nor is it specified whether there was manual review of bibliographic lists.
Results:
The PRISMA figure is not clearly readable; readability should be improved. The same is true for the figures on the risk of study quality.
The table is uncomfortable to read with so many lines cut off, it is recommended to put it horizontally.
Description and analysis are mixed without clear separation.
No comparative analysis is performed between studies (not even systematic qualitative analysis by domains). It is recommended to analyze the results grouped by type of intervention, population or instrument used, for example.
Discussion:
Perhaps they show excessive optimism having found weak evidence, with few experimental studies on the study variable. Many studies not included in the review are cited to reinforce the conclusions, which could lead to confirmation bias.
The contents of the introduction are reiterated without new critical analysis.
Author Response
The work is appropriate for the journal and of interest to readers.
Answer: We acknowledge your comment.
- Title and Abstract:
The title is clear and adequately reflects the content of the study.
The abstract covers the fundamental aspects: justification, method, results and conclusion. However, it uses vague phrases such as “Gratitude appears to be related...” that do not reflect the level of evidence derived from the study. In addition, it mentions 6 included studies but does not distinguish between types of studies (clinical vs. observational trials).
Answer: Thank you for your comment. We corrected these aspects as you suggested (please see section “Abstract”, line 16 and line 19).
- Introduction:
The writing is quite dense and has excessive citations that interrupt the flow.
Some citations are outdated or overused (e.g., Sarzi-Puttini et al., 2020 appears more than 10 times).
Answer: We acknowledge your comment and improved the text, namely by avoiding excessive citations. Regarding the reference “Sarzi-Puttini et al., 2020” we deleted 5 times. However, the is a seminal study published in Nature Reviews in Rheumatology, with 483 citations at the current time.
- Methodology:
The PRISMA 2020 protocol is followed, recorded in PROSPERO and the PICO question is set.
However, the complete search strategy is not detailed, nor is the table of MeSH terms or descriptors provided. Nor is it specified whether there was manual review of bibliographic lists.
Answer: Thank you for your comment. We now provided further detail in lines 149-154 at “Materials and Methods” section.
- Results:
The PRISMA figure is not clearly readable; readability should be improved. The same is true for the figures on the risk of study quality.
The table is uncomfortable to read with so many lines cut off, it is recommended to put it horizontally.
Description and analysis are mixed without clear separation.
No comparative analysis is performed between studies (not even systematic qualitative analysis by domains). It is recommended to analyze the results grouped by type of intervention, population or instrument used, for example.
Answer: Thank you for your recommendations. We meliorate the figures, and we hope that now the figures have more quality. The lack of quality is probably due to the conversion of the word file into pdf. We can also upload the figures separately in the website, so that the final manuscript has a better quality.
We also proceeded as recommended in the table (please see the reconfigured table, now with an horizontal plan).
In the table we organized the studies by chronological order of publication (from the older to the more recent). The different types of intervention and their implementation, the different types of population and the high variety of instruments used make it difficult to present a group analysis since our review only comprised 6 studies. However, throughout the text we referred to the studies in a logical manner and considering the type of study we were talking about (observational vs randomized trial).
- Discussion:
Perhaps they show excessive optimism having found weak evidence, with few experimental studies on the study variable. Many studies not included in the review are cited to reinforce the conclusions, which could lead to confirmation bias.
The contents of the introduction are reiterated without new critical analysis.
Answer: Thank you for your comment. We tempered the “Discussion” and the “Conclusions” sections. Based on question 3 of reviewer 1 the section “Conclusions” were completely changed to provide our proposal about potential study directions and therapeutic interventions for the implementation of gratitude practices in FM management. Please see the section “Conclusions”.
Round 2
Reviewer 2 Report
Comments and Suggestions for Authors
I would like to thank the authors for their efforts in considering all of the reviewer's suggestions. I believe that the work has improved and will benefit the journal and its readers.
Kind regards,